

# Matrilineal phylogeny and habitat suitability of the endangered spotted pond turtle (*Geoclemys hamiltonii*; Testudines: Geoemydidae): a two-dimensional approach to forecasting future conservation consequences

Shantanu Kundu[1], Tanoy Mukherjee[2], Manokaran Kamalakannan[3], Gaurav Barhadiya[4], Chirashree Ghosh[4] and Hyun-Woo Kim[1,5]

[1] Department of Marine Biology, Pukyong National University, Busan, South Korea
[2] Agricultural and Ecological Research Unit, Indian Statistical Institute, Kolkata, West Bengal, India
[3] Mammal and Osteology Section, Zoological Survey of India, Kolkata, West Bengal, India
[4] Department of Environmental Studies, University of Delhi, New Delhi, New Delhi, India
[5] Research Center for Marine Integrated Bionics Technology, Pukyong National University, Busan, South Korea

Corresponding author
Hyun-Woo Kim, kimhw@pknu.ac.kr

## ABSTRACT

The spotted pond turtle (*Geoclemys hamiltonii*) is a threatened and less explored species endemic to Bangladesh, India, Nepal, and Pakistan. To infer structural variation and matrilineal phylogenetic interpretation, the present research decoded the mitogenome of *G. hamiltonii* (16,509 bp) using next-generation sequencing technology. The mitogenome comprises 13 protein-coding genes (PCGs), 22 transfer RNAs (tRNAs), two ribosomal RNAs (rRNAs), and one AT-rich control region (CR) with similar strand symmetry in vertebrates. The ATG was identified as a start codon in most of the PCGs except Cytochrome oxidase subunit 1 (cox1), which started with the GTG codon. The non-coding CR of *G. hamiltonii* was determined to have a unique structure and variation in different domains and stem-loop secondary structure as compared with other Batagurinae species. The PCGs-based Bayesian phylogeny inferred strong monophyletic support for all Batagurinae species and confirmed the sister relationship of *G. hamiltonii* with Pangshura and Batagur taxa. We recommend generating more mitogenomic data for other Batagurinae species to confirm their population structure and evolutionary relationships. In addition, the present study aims to infer the habitat suitability and habitat quality of *G. hamiltonii* in its global distribution, both in the present and future climatic scenarios. We identify that only 58,542 km$^2$ (7.16%) of the total range extent (817,341 km$^2$) is suitable for this species, along with the fragmented habitats in both the eastern and western ranges. Comparative habitat quality assessment suggests the level of patch shape in the western range is higher (71.3%) compared to the eastern range. Our results suggest a massive decline of approximately 65.73% to 70.31% and 70.53% to 75.30% under ssp245 and ssp585 future scenarios, respectively, for the years between 2021–2040 and 2061–2080 compared with the current distribution. The present study indicates that proper conservation management requires greater

attention to the causes and solutions to the fragmented distribution and safeguarding of this endangered species in the Indus, Ganges, and Brahmaputra (IGB) river basins.

# INTRODUCTION

Turtles, terrapins, and tortoises (order Testudines, commonly referred to as turtles) have existed since the Triassic (≈200 million years ago), and approximately 360 extant species are recognized throughout the world (*TTWG (Turtle Taxonomy Working Group), 2021*). Among them, the family Geoemydidae comprises 71 species under three subfamilies (Batagurinae, Geoemydinae, and Rhinoclemmydinae) and 19 genera. The freshwater spotted pond turtle (*Geoclemys hamiltonii*) is medium-sized and classified under the subfamily Batagurinae and the monotypic genus *Geoclemys*. This distinct evolutionary species is distributed in the Indus, Ganges, and Brahmaputra (IGB) river basins in eastern Pakistan, northern India, Bangladesh, and up to northeast India (*Das & Bhupathy, 2010*).

The recent assessment by the International Union for Conservation of Nature (IUCN) Tortoise and Freshwater Turtle Specialist Group (TFTSG) declared *G. hamiltonii* an "endangered" species in the IUCN Red List of Threatened Species (*Praschag, Ahmed & Singh, 2019*), and in Appendix I in CITES (the Convention on International Trade in Endangered Species of Wild Fauna and Flora). This species confronts several threats like habitat destruction, pet trade, and accidental capture by fishing gear throughout its range. Several studies of *G. hamiltonii* have been accomplished to unwrap their distribution, reproduction, and breeding in captivity, conservation status, and systematic revision (*Basu & Singh, 1998*; *Choudhury, Bhupathy & Hanfee, 2000*; *Artner, 2006*; *Ahmed & Das, 2010*; *Das & Bhupathy, 2010*). The conservation status of endangered *G. hamiltonii* is unparalleled throughout its range distribution, demarcated by political boundaries. It is regarded as a "Schedule I" species in the Indian Wildlife (Protection) Act 1972, a "Schedule III" species in the Bangladesh Wildlife (Preservation) Act 1974, and a "Schedule III" species in the Pakistan provincial NWFP Wildlife Act 1975 and the Punjab Wildlife Act 1974. However, to settle their conservation assessment, both molecular and distribution modeling studies across their range can play an important role at this point.

The advancement of molecular tools is unfolding rapidly and has successfully resolved many questions on Geoemydidae turtle systematics (*Praschag, Hundsdörfer & Fritz, 2007*). To date, PCR-based restriction fragment length polymorphism (RFLP) and partial nucleo-mitochondrial gene sequences have been generated for conservation genetics (*Kundu et al., 2018a*; *Chang et al., 2018*; *Rajpoot, Bahuguna & Kumar, 2019*; *Yadav et al., 2021*) and have clarified the phylogenetic position of this turtle group (*Spinks et al., 2004*; *Sasaki et al., 2006*; *Le, McCord & Iverson, 2007*; *Rohilla & Tiwari, 2008*; *Reid et al., 2011*; *Thomson, Spinks & Shaffer, 2021*).

The mitogenomes and phylogenomic data have been largely utilized to interpret the deep evolutionary branching of turtles (*Zardoya & Meyer, 1998*; *Kumazawa & Nishida, 1999*; *Fong et al., 2012*; *Crawford et al., 2015*; *Shaffer et al., 2017*; *Kundu et al., 2018b*). However, complete mitogenomes for representatives of this group, encompassing a large extent of their distribution, are still lacking. Among the Batagurinae subfamily, seven mitogenomes (including the previously generated mitogenome for *G. hamiltonii*) of six species have been generated so far. The genomic features and phylogeny have been elaborated for *Batagur trivittata* (*Feng et al., 2017*), *Pangshura tentoria* (Kundu et al., 2019), and Pangshura sylhetensis (*Kundu et al., 2020*), and the mitogenomes of two *Batagur* turtles (*B. kachuga* and *B. dhongoka*) were also recently analyzed (*Kumar et al., 2021*). Lastly, the mitogenomes of *G. hamiltonii* and *Batagur affinis* have been generated from China (outside range, vouchered at the Turtle Research and Conservation Center of Hainan Normal University) and Malaysia, respectively (animal in captivity), but neither rendered any structural variations or phylogenetic interpretation. Hence, the present study aimed to generate the complete mitogenome of *G. hamiltonii* from the known range distribution in India and perform structural characterization and phylogenetic inferences relative to other Geoemydidae species to obtain more detailed insights on this species evolutionary history.

On the other hand, Testudines conservation status faces the highest anthropogenic pressure among all vertebrates worldwide (*Stanford et al., 2020*). Among the most endangered turtles in the world, the subfamily Batagurinae species are at the top of the list (*Rhodin et al., 2018*). Habitat destruction and fragmentation are the most critical factors that have increased the vulnerability of many freshwater turtles and pushed them to the brink of extinction. Furthermore, many turtle species are vulnerable to climate change because of their restricted dispersion capacities and extensive temperature-dependent sex determination, which has increased dramatically over the last decade (*Butler, 2019*; *Mothes, Howell & Searcy, 2020*; *Willey et al., 2022*).

In this context, species distribution modeling (SDM) has the potential to predict relevant information regarding the present habitat condition with high precision by using prior information about the species and associated ecological envelope across space and time (*Guisan & Zimmermann, 2000*; *Elith & Leathwick, 2009*). The SDM remains a key as it helps in finding ecological and biogeographical relationships for developing conservation and management strategies (*Peterson, 2007*; *Guisan et al., 2013*). In recent years, the incorporation of ecophysiological models has been critical in SDM projections of many vertebrate species in order to comprehend range shifts in response to climate change (*McMahon et al., 2011*; *Bellard et al., 2012*; *Ceia-Hasse et al., 2014*; *Murali et al., 2023*). Hence, the present study was further intended to execute a different dimension to visualize the spatial features of ecological hypervolume as well as the present and future habitat projections of *G. hamiltonii* in the IUCN range.

Such a two-dimensional approach with genetic and species distribution modeling information will help researchers and conservation practitioners develop better-informed management and action plans for the benefit of *G. hamiltonii* in India and neighboring countries. The current study further stimulates long-term monitoring of *G. hamiltonii* by
the IUCN-TFTSG and Turtle Survival Alliance (TSA) to protect wild populations in its native range.

# MATERIALS AND METHODS

## Species identification and sampling

The unique specimen (adult male) utilized in this study was detected in New Delhi, India (28.51N, 77.20E), which was used for ornamental commercial purposes and identified as *Geoclemys hamiltonii* by the key characters (*Das & Bhupathy, 2010*) (Fig. 1). The organism was sedated by using 20–30 mg/kg Alfaxolone SC, and a small amount of blood sample (100 µl) was collected from the hind limb with sufficient care and preserved in an EDTA-containing 1.5 ml centrifuge tube at 4 °C. No animals were collected from the wild or specimen vouchered in the present study. Therefore, this scientific research is not concerned with animal ethics issues, and does not require ethics committee or institutional review board approval. The experimental protocols were approved by the host institutions (Pukyong National University, South Korea; Zoological Survey of India; Indian Statistical Institute; and University of Delhi, India), and all procedures and representations were accomplished in accordance with relevant guidelines and regulations of ARRIVE 2.0. (https://arriveguidelines.org) (*Percie du Sert et al., 2020*).

## Mitochondrial DNA extraction, sequencing, and mitogenome assembly

The molecular experiments, mitogenome sequencing, and assembly were executed at Unipath Specialty Laboratory Ltd. (http://www.unipath.in/), India. The mitochondrial DNA was extracted by using Alexgen DNA kit (Alexius Biosciences, Ahmedabad, Gujarat, India) followed by the published protocol (*Ahmad et al., 2007*), and the quantity was measured using a Qubit®4.0 fluorometer.

The paired-end sequencing library was developed using the QIAseq FX DNA Library Kit (CAT-180479). DNA was mechanically sheared into smaller fragments by the Covaris M220 Focused Ultrasonicator (Covaris Inc., San Diego, CA, USA), and illumine-specific adapters were ligated to both ends of the DNA fragments. To assure maximum yields from restricted quantities of starting material, the HiFi PCR Master Mix (Takara Bio Inc., Kusatsu, Shiga, Japan) was used to perform a high-fidelity amplification step. The amplified libraries were analyzed on TapeStation 4150 (Agilent Technologies, Santa Clara, CA, USA) by using High Sensitivity D1000 ScreenTape® as per the manufacturer's protocols. The library was loaded onto the Illumina Novaseq 6000 platform for cluster generation and sequencing (Illumina, San Diego, CA, USA) after getting the qubit concentration and the mean peak size from the tape station profile. The high-quality paired-end reads were assembled and annotated using NOVOPlasty v. 4.2 (*Dierckxsens, Mardulyn & Smits, 2017*).

## Mitogenome characterization and phylogenetic analyses

The boundaries and strand directions of each gene were affirmed by the MITOS v806 online webserver (http://mitos.bioinf.uni-leipzig.de) (*Bernt et al., 2013*).

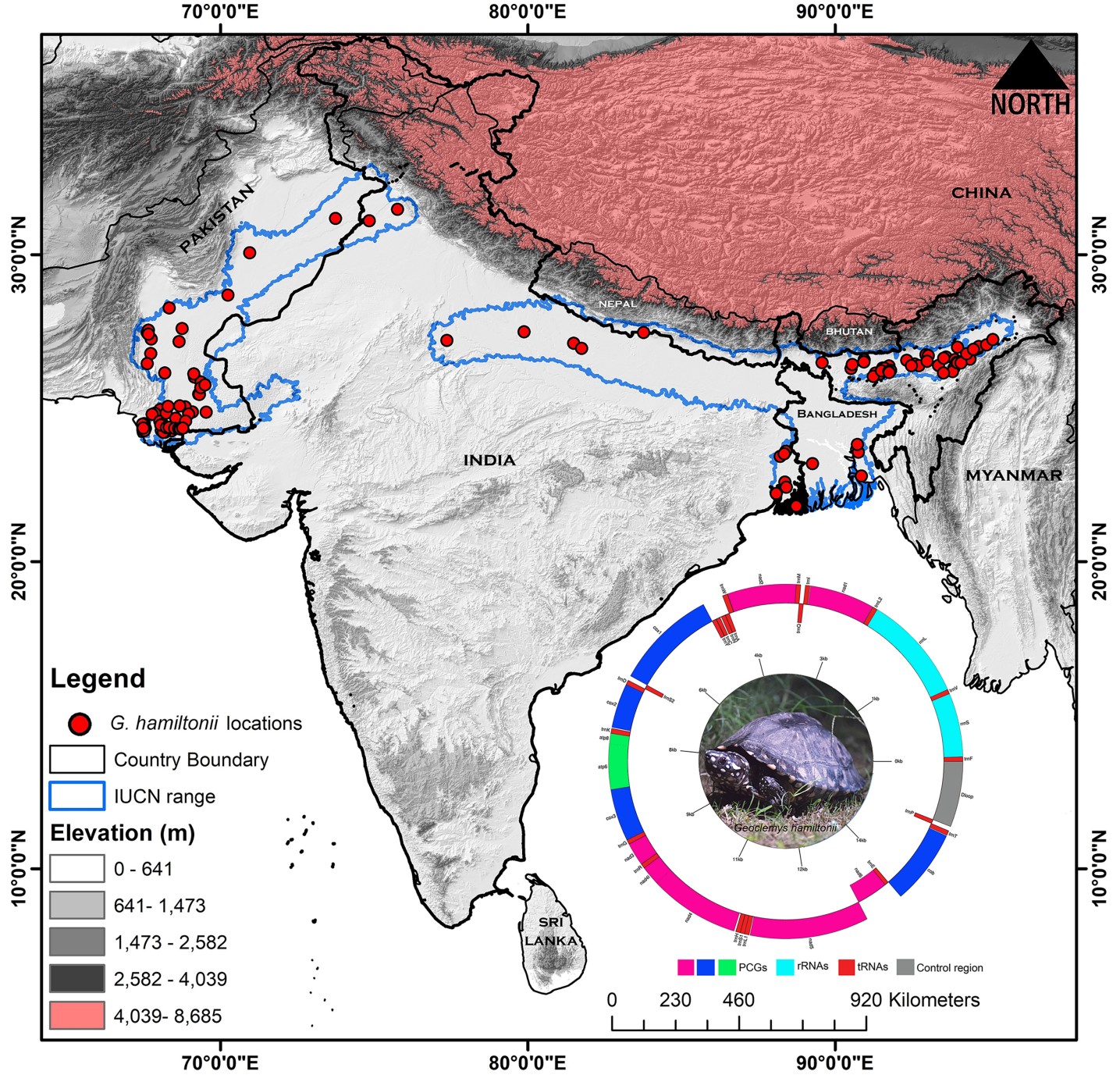

**Figure 1** **Map displaying the global range distribution of *Geoclemys hamiltonii* marked by a blue line.** The map was prepared by ArcGIS 10.6 using polygons (.shp file) acquired from the IUCN Red List of Threatened Species (assessed on 20 May 2023). The locations of *G. hamiltonii* were obtained from previous literature and GBIF online data repository (assessed on 20 May 2023) and marked by red dots. The embedded mitochondrial genome of *G. hamiltonii* with gene boundaries (plotted by GenomeVX webserver). The species photograph was taken by Gaurav Barhadiya. Tanoy Mukherjee prepared the map using ArcGIS 10.6 and edited it with Adobe Photoshop CS 8.0.      

The protein-coding genes (PCGs) were validated after assuring the putative amino acid sequences of vertebrate mitochondrial genetic code through the ORF Finder web tool (https://www.ncbi.nlm.nih.gov/orffinder/), and initiation, as well as termination codons,

were identified by the reference mitochondrial genome (accession number ON243873). The generated mitogenome was submitted to GenBank through the Sequin submission tool. The circular illustration of the *G. hamiltonii* mitogenome was plotted using the GenomeVX webserver (http://wolfe.ucd.ie/GenomeVx/) (*Conant & Wolfe, 2008*). The intergenic spacers and overlapping regions between the neighboring genes were labeled manually.

The size and nucleotide composition of each gene were estimated using MEGA11 (*Tamura, Stecher & Kumar, 2021*). The base composition skew was calculated as described: AT-skew = [A − T]/[A + T]; GC-skew = [G − C]/[G + C] in the previous study (*Perna & Kocher, 1995*). The ribosomal RNA gene (rRNA) and transfer RNA gene (tRNA) boundaries of *G. hamiltonii* were also affirmed through the MITOS online server. To determine the structural domains and putative secondary structures, the control region (CR) of *G. hamiltonii* was visualized through the Mfold web server (http://unafold.rna.albany.edu) and Vienna RNA package (https://www.tbi.univie.ac.at/RNA/) (*Zuker, 2003*; *Hofacker, 2003*) and compared with other Batagurinae species manually. The online Tandem Repeats Finder web tool (https://tandem.bu.edu/trf/trf.html) was used to predict the tandem repeats in the CR (*Benson, 1999*).

To assess the evolutionary relationships, a total of 42 Geoemydinae species mitogenomes were acquired from GenBank (Table S1). The Asian Forest tortoise, *Manouria emys* (family Testudinidae), mitogenome was used as an outgroup in the present analysis. To construct the dataset for phylogenetic analysis, the PCGs were discretely aligned in TranslatorX with the MAFFT algorithm and the L-INS-i approach with GBlocks parameters (*Abascal, Zardoya & Telford, 2010*) and concatenated by SequenceMatrix v1.7.84537 (*Vaidya, Lohman & Meier, 2010*). The finest model was computed by partitioning each PCG using PartitionFinder 2 (*Lanfear et al., 2016*) at the CIPRES Science Gateway V. 3.3 (*Miller et al., 2015*). The Bayesian (BA) tree was constructed with Mr. Bayes 3.1.2 (*Ronquist & Huelsenbeck, 2003*) by choosing nst = 6, one cold and three hot Metropolis-coupled Markov chain Monte Carlo (MCMC), and it was run for 1,000,000 generations with tree sampling at every 100th generation, with 25% of samples rejected as burn-in. Further, the Maximum-Likelihood (ML) tree was constructed by using the W-IQ-TREE web server (http://iqtree.cibiv.univie.ac.at/) (*Trifinopoulos et al., 2016*) with 1,000 bootstrap replications. Both BA and ML topologies were further processed in iTOL v4 (https://itol.embl.de/login.cgi) for better visualization (*Letunic & Bork, 2007*).

## Species occurrence information

The extent range boundary of *G. hamiltonii* range distribution was downloaded from the IUCN (https://www.iucnredlist.org/) and the map was built by ArcGIS 10.6 software (ESRI1, Redlands, CA, USA) (Fig. 1). The occurrence records of *G. hamiltonii* were collected from previous literature (Table S2) and the Global Biodiversity Information Facility (GBIF) online repository system (https://doi.org/10.15468/dl.ce6mmr) (*GBIF, 2023*). We collected (*n* = 136) spatially independent occurrence points for *G. hamiltonii*, which are adequate for the distribution modeling of the targeted species (*Wieczorek, Guo & Hijmans, 2004*; *Bloom, Flower & DeChaine, 2018*) (Fig. 1). Spatial autocorrelation was

executed by using the SDM Toolbox on the locality points with a search radius of 1 km based on the raster resolution of the predictor variables to minimize the overfitting of the model (*Brown, 2014*).

## Model covariate selection

Considering the ecological requirements of *G. hamiltonii*, the variables that may play a substantial role in predicting the suitable habitat were preferred for primary screening (*Peterson et al., 2011*). We selected 25 habitat variables and sorted them into four types: topographic, land cover and land use (LCLU), climatic, and anthropogenic (Table S3). The climatic conditions corresponded to the standard 19 bioclimatic variables from Worldclim, Version 2.0 (https://www.worldclim.org/) (*Fick & Hijmans, 2017*).

To examine the effect of individual LCLU classes, land use and land cover derived from Copernicus Global Land Service (https://lcviewer.vito.be/download) were used (*Buchhorn et al., 2020*). The Global Human Footprint Dataset was used as an anthropogenic forecaster to provide entropy on the Human Influence Index (HII) to better comprehend human influence on target species (*SEDAC, 2005*). We utilized water occurrence intensity and distance to major water bodies (https://www.diva-gis.org/gdata) to assess the influence of water availability and aquatic preference on the species (*Pekel et al., 2016*), which were calculated by using the Euclidian distance function in ArcGIS 10.6.

The topographic variables, such as slope and elevation, were yielded using the 90-m Shuttle Radar Topography Mission (SRTM) data (http://srtm.csi.cgiar.org/srtmdata/). All predictors were resampled at 1 km² spatial resolution using the spatial analysis tool within ArcGIS 10.6. The spatial multicollinearity within the predictors was screened using SDM Toolbox v2.4, and the variables with $r > 0.8$ Pearson's correlation were removed from the final model (*Warren, Glor & Turelli, 2010*).

Furthermore, for climate change projections in two different SSP (Shared Socio-economic Pathways), *i.e.*, ssp245 and ssp585, future scenarios for the years between 2021–2040 and 2061–2080, we have used the General Circulation Model (GCM) developed by the Beijing Climate Centre (BCC) BCC-CSM 2 MR (*Xin et al., 2018*). To evaluate the effect of climate change for the present study, we have kept the non-climatic raster constant.

## Model building and evaluation

Due to its high performance in predicting species distribution models, we used MaxEnt Ver. 3.4.4, which is known to execute well even when the number of covariates exceeds the number of occurrences for a predictive model (*Phillips & Dudík, 2008*; *Peacock, 2011*; *Erinjery, Singh & Kent, 2019*). Further, we used the bootstrapping replication approach and Bernoulli generalized linear model with the ClogLog link function for developing the present model (*Phillips et al., 2017*). The model utilized the training data on each occurrence point as n-1 and examined the model execution with the residual points and 50 runs as replicates (*Elith et al., 2011*; *Peacock, 2011*). The results generate a probability distribution outcome as an uninterrupted probability surface raster of the analysis extent

ranging from 0–1, with '1' as the most suitable habitat and '0' being the least suitable habitat for *G. hamiltonii*.

Variable influence on the occurrences was estimated using the Jackknife test of acquired regularized training gain (*Phillips & Dudık, 2008*). For model evaluation, we used the area under the curve statistics (AUC) of the receiver operating characteristic (ROC) curves (*Halvorsen et al., 2016*). The AUC test statistic value ranges from 0 to 1, where values lower than 0.5 indicate deficient power; minimum discrimination among the predictive presence and absent areas is considered to be deficient; 0.5 indicates a random prediction; 0.7–0.8 is regarded as an acceptable model result; 0.8-0.9 is considered to be excellent; and <0.9 is regarded as an exceptional model (*Kamilar & Tecot, 2016*; *Johnson et al., 2016*). The binary maps were made based on an equal test sensitivity and specificity (SES) threshold for the predicted suitable habitat for the targeted species and used the raster calculator to evaluate the zonal statistics using the Zonal Statistics Tool in ArcGIS 10.6.

## Assessment of habitat quality

The comparative analyses were performed between the suitable areas of the eastern and western ranges of *G. hamiltonii* for both the present and future climatic models. We used FRAGSTATS version 4.2.1 to estimate the class level metrics, *i.e.*, number of patches (NP), aggregate index (AI), patch density (PD), largest patch index (LPI), edge density (ED), total edge (TE), and landscape shape index (LSI), as the indices of the level of habitat character and level of fragmentation indicators in the modeled area for present and climatic change scenarios (*McGarigal, 2015*; *Mukherjee et al., 2020*; *Xia et al., 2020*).

# RESULTS

## Mitogenome characterization and comparison

The mitogenome sequences of 35 Geoemydinae species, six Batagurinae species, and one Rhinoclemmydinae species have been assembled so far (https://www.ncbi.nlm.nih.gov/genome/organelle/). The present study characterizes the mitogenome sequence of *Geoclemys hamiltonii* to elucidate its evolutionary significance in the Testudines-tree of life. The mitogenome (16,509 bp) of the spotted pond turtle, *G. hamiltonii* was determined (GenBank accession number OP344485). The circular mitogenome consists of 13 protein-coding genes (PCGs), 22 transfer RNA genes (tRNAs), two ribosomal RNA genes (rRNAs), and a major non-coding AT-rich control region (CR). Among them, nine genes (*nad6* and eight tRNAs) were located on the light strand, while the other 28 genes were located on the heavy strand (Fig. 1, Table 1). Across the Batagurinae subfamily, the length of the mitogenomes varied from 16,505 bp (*G. hamiltonii*, generated from China) to 16,657 bp (*P. tentoria*). All Batagurinae turtles displayed strand symmetry as detected in typical vertebrates mitogenomes (*Anderson et al., 1982*).

The structural features of both mitogenomes (India and China) are almost similar. The nucleotide composition of the *G. hamiltonii* mitogenomes generated from India (OP344485) and China (ON243873) was A+T biased at 59.47% and 59.44%, respectively. A total of seven base-pair variable sites were identified in both mitogenomes of *G. hamiltonii*. The AT skew and GC skew were 0.13 and −0.34 in both mitogenomes of

**Table 1 List of annotated mitochondrial genes of *Geoclemys hamiltonii*.**

| Gene | Direction | Location | Size | Anti-codon | Start codon | Stop codon | Intergenic nucleotides |
|------|-----------|----------|------|------------|-------------|------------|------------------------|
| *trnF* | + | 1–70 | 70 | TTC | . | . | 0 |
| *rrnS* | + | 71–1,029 | 959 | . | . | . | 0 |
| *trnV* | + | 1,030–1,100 | 71 | GTA | . | . | 0 |
| *rrnL* | + | 1,101–2,704 | 1,604 | . | . | . | 0 |
| *trnL2* | + | 2,705–2,780 | 76 | TTA | . | . | 0 |
| *nad1* | + | 2,781–3,748 | 968 | . | ATG | TAA | 0 |
| *trnI* | + | 3,749–3,818 | 70 | ATC | . | . | −1 |
| *trnQ* | − | 3,818–3,888 | 71 | CAA | . | . | −1 |
| *trnM* | + | 3,888–3,956 | 69 | ATG | . | . | 0 |
| *nad2* | + | 3,957–4,995 | 1,039 | . | ATG | TAA | 0 |
| *trnW* | + | 4,996–5,071 | 76 | TGA | . | . | 1 |
| *trnA* | − | 5,073–5,141 | 69 | GCA | . | . | 1 |
| *trnN* | − | 5,143–5,215 | 73 | AAC | . | . | 25 |
| *trnC* | − | 5,241–5,306 | 66 | TGC | . | . | 5 |
| *trnY* | − | 5,312–5,382 | 71 | TAC | . | . | 1 |
| *cox1* | + | 5,384–6,934 | 1,551 | . | GTG | AGG | −12 |
| *trnS2* | − | 6,923–6,993 | 71 | TCA | . | . | 0 |
| *trnD* | + | 6,994–7,063 | 70 | GAC | . | . | 0 |
| *cox2* | + | 7,064–7,750 | 687 | . | ATG | TAG | 0 |
| *trnK* | + | 7,751–7,825 | 75 | AAA | . | . | 1 |
| *atp8* | + | 7,827–7,994 | 168 | . | ATG | TAA | −10 |
| *atp6* | + | 7,985–8,668 | 684 | . | ATG | TAA | −1 |
| *cox3* | + | 8,668–9,451 | 784 | . | ATG | TAA | 0 |
| *trnG* | + | 9,452–9,519 | 68 | GGA | . | . | 1 |
| *nad3* | + | 9,521–9,853 | 333 | . | ATG | AGG | 17 |
| *trnR* | + | 9,871–9,940 | 70 | CGA | . | . | 0 |
| *nad4l* | + | 9,941–10,237 | 297 | . | ATG | TAA | −7 |
| *nad4* | + | 10,231–11,607 | 1,377 | . | ATG | TAA | 15 |
| *trnH* | + | 11,623–11,692 | 70 | CAC | . | . | 0 |
| *trnS1* | + | 11,693–11,758 | 66 | AGC | . | . | −1 |
| *trnL1* | + | 11,758–11,829 | 72 | CTA | . | . | 0 |
| *nad5* | + | 11,830–13,638 | 1,809 | . | ATG | TAA | −5 |
| *nad6* | − | 13,634–14,161 | 528 | . | ATG | AGG | 0 |
| *trnE* | − | 14,162–14,229 | 68 | GAA | . | . | 5 |
| *cytb* | + | 14,235–15,378 | 1,144 | . | ATG | TAA | 0 |
| *trnT* | + | 15,379–15,450 | 72 | ACA | . | . | 1 |
| *trnP* | − | 15,452–15,520 | 69 | CCA | . | . | 0 |
| CR | . | 15,521–16,509 | 989 | . | . | . | . |

*G. hamiltonii, respectively.* A total of eight overlapping regions (total length of 38 bp) were identified in both *G. hamiltonii* mitogenomes, with the longest region (12 bp) between cytochrome oxidase subunit 1 (*cox1*) and tRNA-serine (*trnS2*). Further, a total
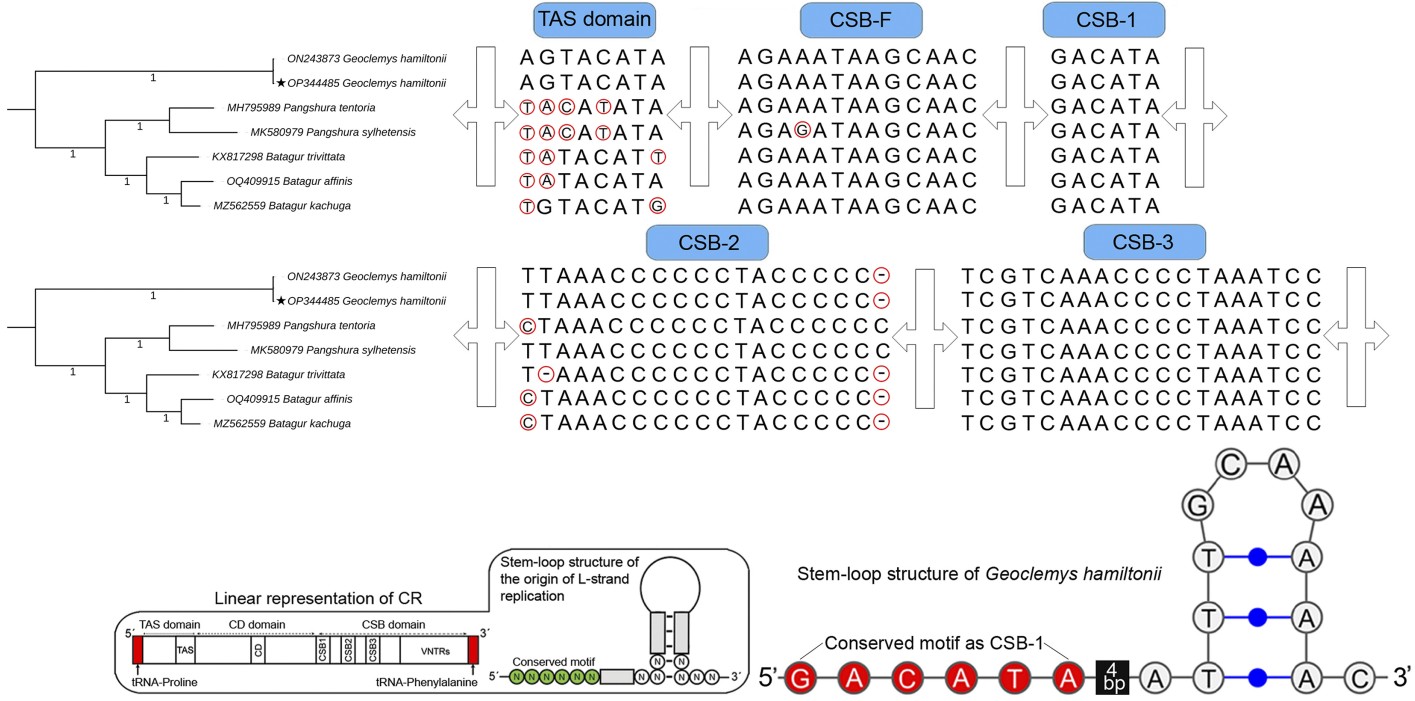

**Figure 2 Structural variation within the different domains of *G. hamiltonii* control region compared with other Batagurinae species.** The linear representation and stem-loop structure of the origin of L-strand replication anticipated by the Mfold web server (http://unafold.rna.albany.edu) and merged manually on the map by Adobe Photoshop CS 8.0. The accession number with a star indicates the sequence generated from India.

of 11 intergenic spacer regions (total length of 73 bp) were also found in both *G. hamiltonii* mitogenomes, with the longest region (25 bp) between tRNA-asparagine (*trnN*) and tRNA-cysteine (*trnC*), which acts as the origin of L-strand replication. The total length of PCGs was 11,369 bp (68.88%); rRNAs were 2,563 bp (15.53%); tRNAs were 1,588 bp (9.62%) and 1,553 bp (9.41%); and CR was 989 bp (5.99%) and 985 bp (5.97%) in both mitogenomes generated from India and China, respectively. Most of the PCGs of *G. hamiltonii* mitogenomes started with the ATG codon; however, the GTG initiation codon was observed in the *cox1* gene. The AGG termination codon was used by *cox1*, *nad3*, and *nad6*; TAG by *cox2*; and TAA by *atp8*, *atp6*, *nad4l*, and *nad5*. The incomplete TAA termination codon was detected in five PCGs (*nad1*, *nad2*, *cox3*, *nad4*, and *cytb*). Among the 22 tRNA genes in both mitogenomes, 14 were found on the majority strand, and the remaining eight genes were on the light strand with specific anticodons.

The total length of *G. hamiltonii* CR was 989 bp (India) and 985 bp (China), within the range of 947 bp (*B. trivittata*) and 1,151 bp (*P. tentoria*) in other Batagurinae species. The CR of *G. hamiltonii* was also typically constructed with three functional domains: the termination associated sequence (TAS), the conserved sequence block (CSB), and the central conserved (CD), as illustrated in other Testunines (*Bernacki & Kilpatrick, 2020*). Species-specific structural variations were observed in the TAS, CSB-F, and CSB-2 domains (Fig. 2). The CR of *G. hamiltonii* is also implied in the initiation of replication and is placed between tRNA-proline and tRNA-phenylalanine, as depicted in most of the

Testudines. In *G. hamiltonii*, 4 bp gaps were present between CSB-1 and the stem loop, which could be used as species-specific markers. A total of 44.5 times of two base pairs (TA) tandem repeats were encountered in the VNTRs (variable number tandem repeats) region in the generated *G. hamiltonii* non-coding CR, whereas the other mitogenome generated from China (outside range) revealed 42.5 times of two base pairs (TA), 3.9 times 23 bp, and 3.1 times 24 bp repeats.

## Major phylogenetic relationship

The evolutionary relationship, origin, and diversification of turtles and tortoises have been assessed in the last few decades (*Barley et al., 2010*; *Crawford et al., 2015*; *Shaffer et al., 2017*), including a comprehensive phylogeny of all extant Testudines species that relates their diversity with historical climate shifts on the continental margins of the earth (*Thomson, Spinks & Shaffer, 2021*). Both conventional and molecular taxonomy have demonstrated the separate lineage of *G. hamiltonii* from other Batagurinae species (*Spinks et al., 2004*; *Le, McCord & Iverson, 2007*; *Thomson, Spinks & Shaffer, 2021*). The mitogenomic data has been effectively employed to infer the evolutionary relationships of many Testudines species, adding the members of the Batagurinae subfamily (*Feng et al., 2017*; *Kundu et al., 2019*, *2020*; *Kumar et al., 2021*).

Both BA and ML phylogenies clearly segregated all the Testudines species, including *G. hamiltonii*, with high posterior probability support (Figs. 3 and S1). The current mitogenomic phylogeny with a combination of 13 PCGs infers a robust phylogeny and supports the sister relationship of *G. hamiltonii* with *Pangshura* and *Batagur* species, as evidenced in previous studies (*Thomson, Spinks & Shaffer, 2021*). The species of the subfamily Geoemydinae displayed paraphyletic clustering in the current mitogenomic dataset, as shown in the most recent research (*Thomson, Spinks & Shaffer, 2021*). Further, the species of the subfamily Rhinoclemmydinae, *Rhinoclemmys punctularia* showed close clustering with two Geoemydinae species (*Geoemyda spengleri* and *Geoemyda japonica*) in the present topology (Fig. 3). The authors recommend the addition of mitogenome data for other Batagurinae species (*Hardella*, *Malayemys*, *Morenia*, and *Orlitia*) from their range area to ensure a comprehensive mitogenomic phylogeny.

This reference sequence obtained in this study will be helpful for further population genetics studies of this endangered species by examining mitochondrial genes. Although several studies with mitochondrial and nuclear markers have been completed to elucidate significant effects on the Geoemydidae phylogeny, this is likely incomplete due to lineage sorting. Thus, the strategy required to address this issue is to add more linked markers from the nuclear genomes and whole genome data of all extant species to address complete phylogenetic relationships.

## Model execution and habitat suitability

The present model precisely predicted the suitable habitats for *G. hamiltonii* within the studied landscape (Fig. 4). The average training AUC for replicate runs for the model was found to be 0.902 ± 0.016 (SD) (Fig. 5). From the total distribution range extent (817,341 km$^2$), about 58,542 km$^2$ (7.16%) is suitable for *G. hamiltonii* (Figs. 4 and S4).

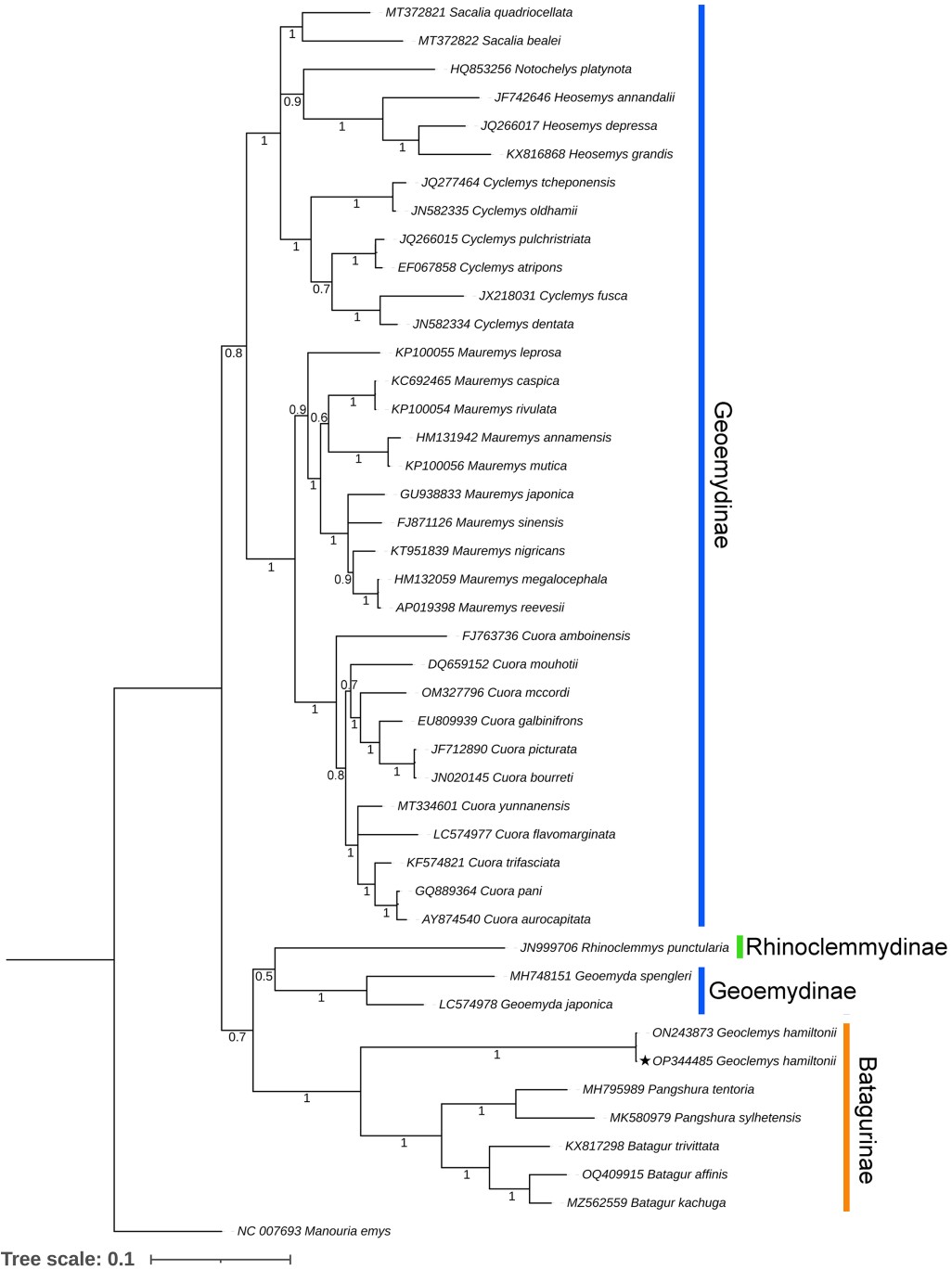

**Figure 3 Unified Bayesian (BA) phylogenetic tree based on the concatenated DNA sequences of 13 PCGs of 42 Geoemydidae species elucidating the evolutionary relationship and placement of *G. hamiltonii*.** The mitogenome of *M. emys* (family Testudinidae) was used as an out-group taxon. The BA posterior probability support of each node was superimposed. The topology was prepared by Mr. Bayes 3.1.2 software and illustrated by the iTOL v4 online server (https://itol.embl.de/login.cgi).

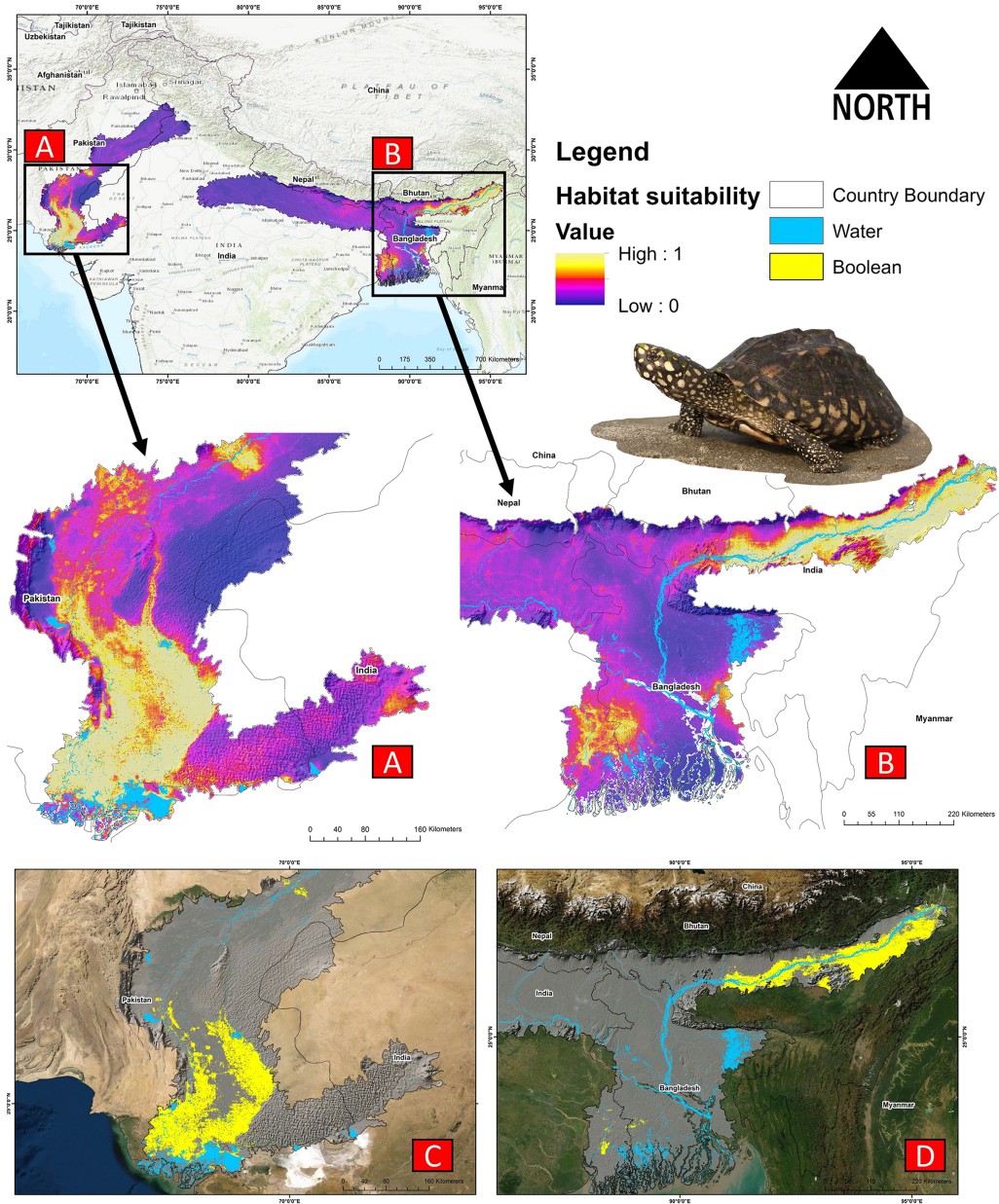

**Figure 4 Representing the likelihood of suitable habitats for *G. hamiltonii* in both the Western and Eastern ranges.** (A) Distribution of suitable areas in the western range. (B) Distribution of Suitable areas in the eastern range. (C) The suitable binary area in the western range. (D) The suitable binary area in the eastern range. All the maps were prepared using ArcGIS 10.6 in the present study. The species photograph was acquired from the free repository Wikimedia Commons (photo taken by Rohit Naniwadekar at Biswanath Ghat, Assam, India) and attributed under Creative Commons Attribution-Share Alike 4.0 International (https://commons.wikimedia.org/wiki/File:Geoclemys_hamiltonii_Biswanath_01.jpg).

The present results also depict distinct habitat patches with fragmented habitats in both the eastern and western ranges. The most suitable areas within the southern range are situated in the far western part (29,872 km$^2$), covering the southern portion of Pakistan (Fig. 4). Further, in the eastern range, the most suitable and unfragmented habitat patches

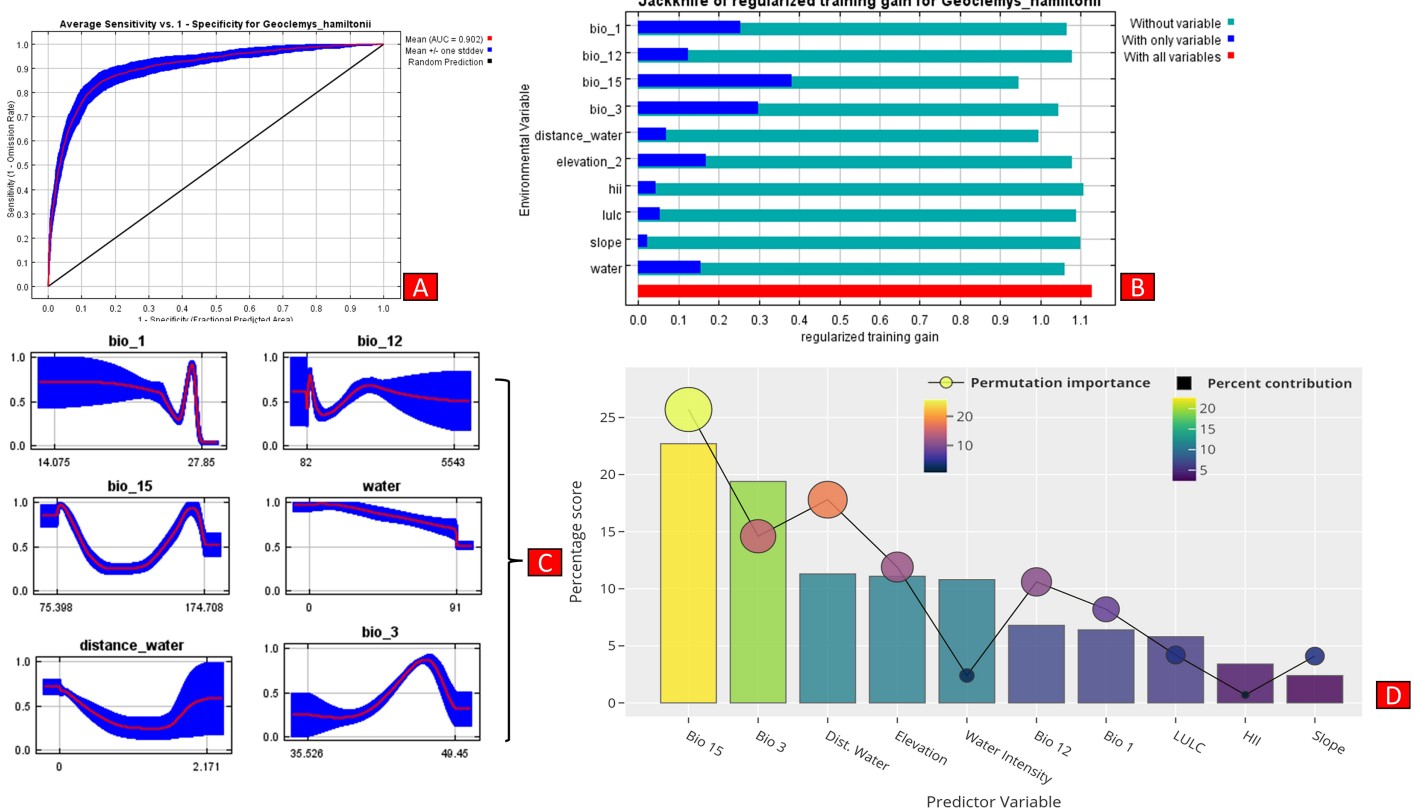

**Figure 5 Showing model evaluation along with variable influence.** (A) The average training ROC (Receiver Operating Characteristics) for the model. (B) Jackknife test for all the selected variables, where blue bar = shows each variable importance in explaining the data variation when used separately. Green bar = showing the loss in overall gain after the particular variable was dropped. Red bar = total model gain. (C) The response curves of the critical predictors governing the habitat suitability of *G. hamiltonii*. (D) The contribution percentage represented by column graph (color ramp represents the %contribution) and permutation importance represented by the circular plot (permutation importance was illustrated by size and color ramp).

(28,670 km$^2$) were demarcated in the far eastern portion of Assam, encompassing the Brahmaputra River. The model indicates that the distribution of habitat patches for *G. hamiltonii* was strongly shaped by precipitation seasonality (coefficient of variation) (Bio 15) with a relative share of 22.7%, followed by the share of isothermality (Bio 3) of 19% (Fig. 5). Further, distance to water bodies (distance water) and water availability (water) were also positively determinants of the distribution of *G. hamiltonii* with percentage contributions of 11.3% and 10.8%, respectively (Fig. 5 and Fig. S3).

The comparative analysis of present and future models suggests a massive decline of approximately 65.73% (ssp245) and 70.53% (ssp585) in future scenarios for the years between 2021 and 2040 (Fig. 6) compared with the current distribution. Furthermore, for the years between 2061 and 2080, the result suggests a decline of 70.31% (ssp245) and 75.30% (ssp585). The area of the most suitable habitats for *G. hamiltonii* was found to be 58,542 km$^2$ in the present scenario. In contrast, in a climatic scenario, it may be reduced to 20,059 and 17,249 km$^2$ at ssp245 and ssp585, respectively, for the year 2040, which can be further reduced up to 14,456 km$^2$ in the year 2080.
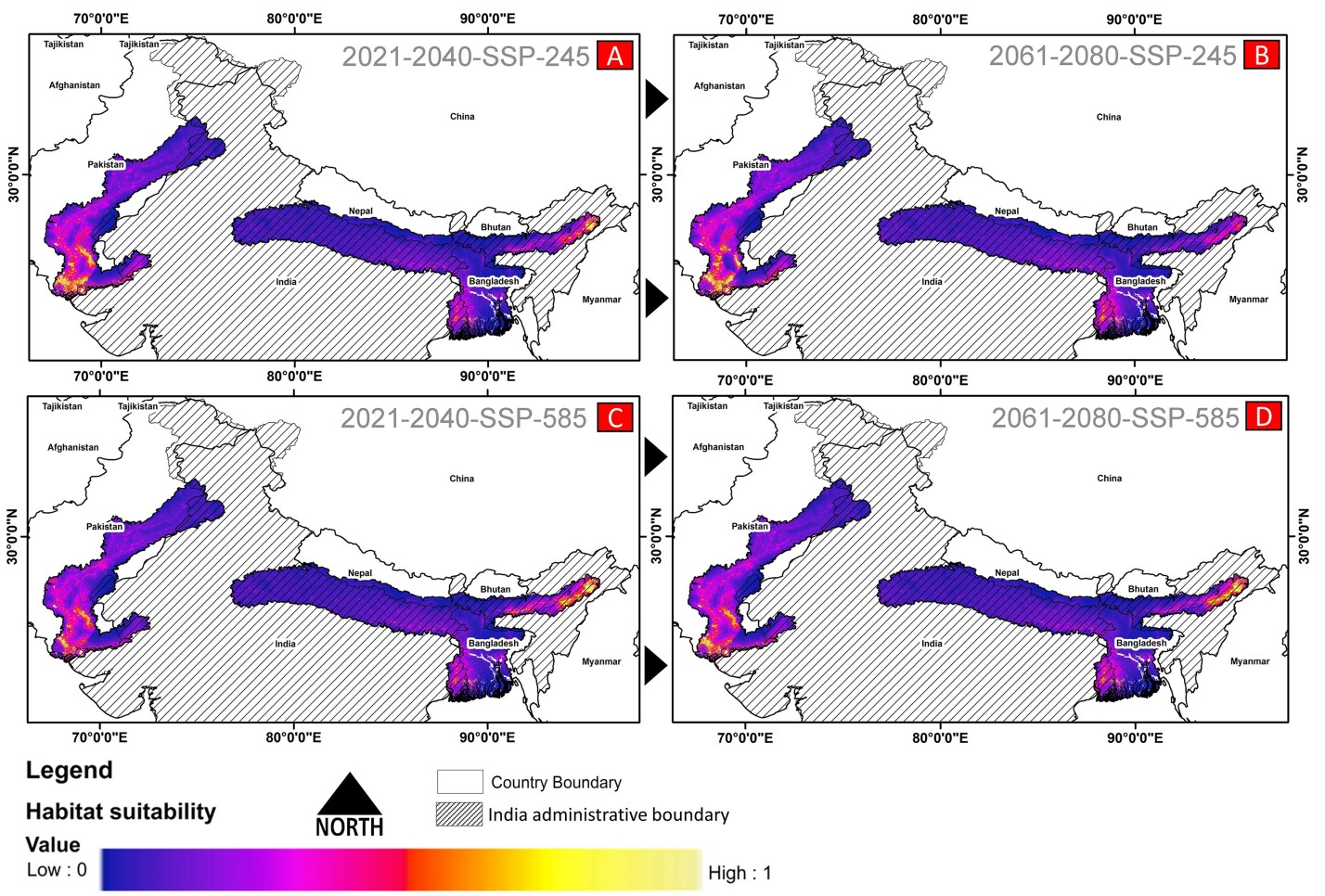

**Figure 6 The habitat suitability for *G. hamiltonii* in future climatic projection scenarios of ssp245 and ssp585 future scenarios for the year 2021–2040 and 2061–2080.** (A) The projection for the years 2021–2040-SSP-245, (B) the year 2021–2040-SSP-585, (C) years 2061–2080-SSP-245, and (D) years 2061–2080-SSP-585. All the maps were prepared using ArcGIS 10.6 in the present study.

## Habitat quality assessment

Higher values of NP, PD, and LPI within the western range were detected, suggesting the presence of multiple larger habitat patches compared to the eastern range (Fig. 7). However, the comparatively higher scores of TE, ED, and AI showed more dispersed and fragmented patches of suitability in the western range. Moreover, the higher aggregation value (86.60) followed by lower values of ED (34,477.52) within the habitat patches in the east range indicates a higher level of habitat integrity among the suitable patches (Fig. 7). The level of patch shape complexity denoted by the LPI for the west range (7.90) has increased by 71.3% compared to the eastern range (4.61), which indicates the level of structural continuity in *G. hamiltonii* habitat in the eastern range compared to western habitat patches (Fig. 7). The current results from the future projections in multiple climate change scenarios suggest a substantial decline in the overall habitat quality in both the eastern and western ranges. The major changes have been signified by a sharp decline in LPI from 7.9 in the western and 4.61 in the eastern ranges to as low as 0.04 in the eastern
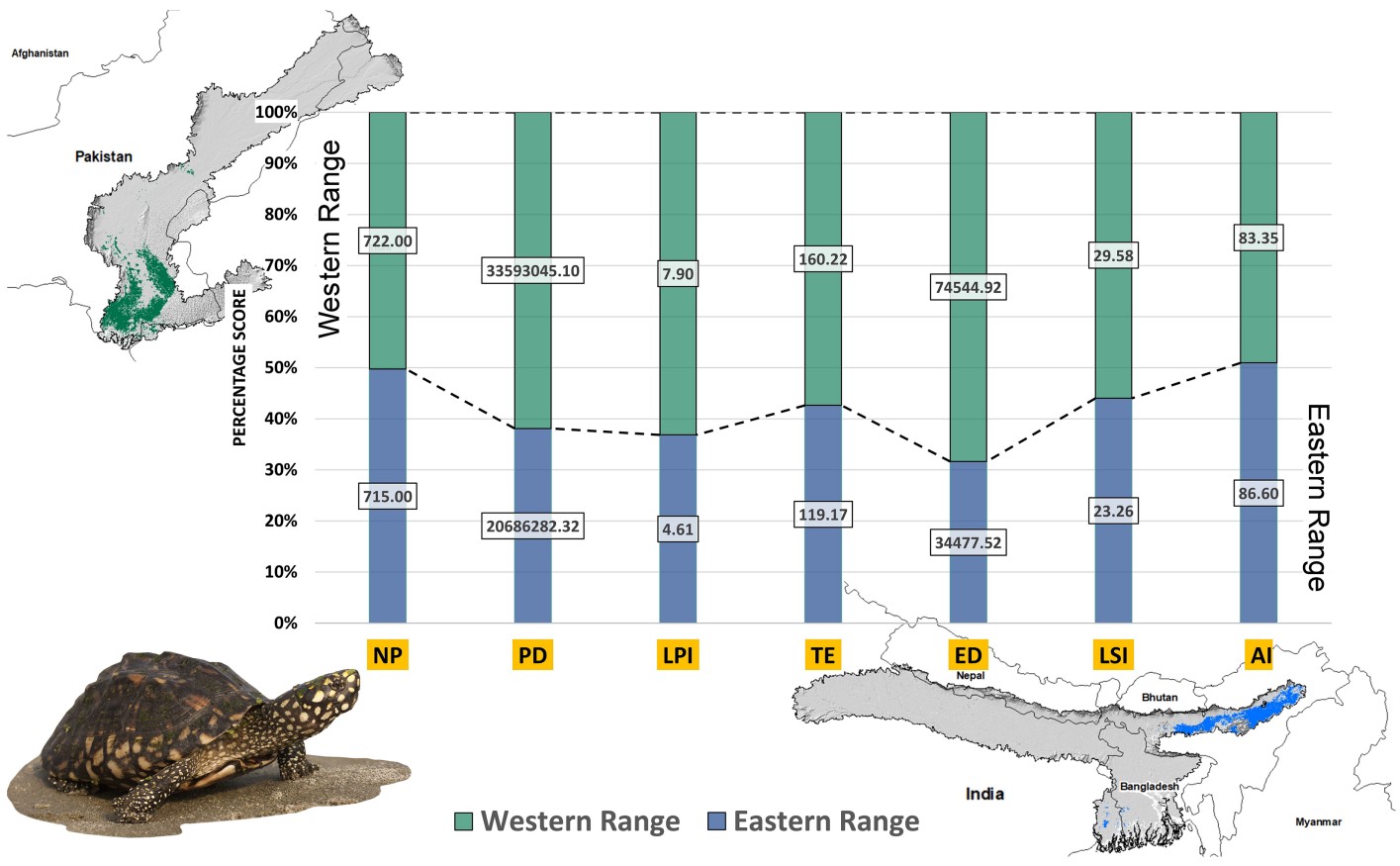

**Figure 7** **The percentage stack of class-level matrices applied for habitat quality assessment of *G. hamiltonii* in the western range (Green) and eastern range (Blue).** Values represent the score of the indices. (NP, No. of Patches; PD, Patch Density; LPI, Largest Patch Index; TE, Total Edge; ED, Edge Density; LSI, Landscape Shape Index; AI, Aggregation Index). All the maps were prepared using ArcGIS 10.6 in the present study. The species photograph was acquired from the free repository Wikimedia Commons (photo taken by Rohit Naniwadekar at Biswanath Ghat, Assam, India) and attributed under Creative Commons Attribution-Share Alike 4.0 International (https://commons.wikimedia.org/wiki/File:Geoclemys_hamiltonii_Biswanath_01.jpg).

and 1.29 in the western ranges for the year 2080, ssp585. Furthermore, the patch aggregation value represented by AI also showed a major reduction in eastern habitat for *G. hamiltonii* by 62.41% ssp585 (Fig. 8).

## DISCUSSION

Over the past 100 years, species extinction rates have increased dramatically and life on earth is currently facing a sixth mass extinction driven by anthropogenic activity, climate change, and ecological collapse (*Teixeira & Huber, 2021*). Hence, protecting biodiversity is a priority to support ecosystems and human well-being with a new unifying concept and the implementation of worthy conservation strategies (*Conde et al., 2019*).

The present study assembled and characterized the mitogenome of *G. hamiltonii* from India and confirmed the evolutionary dynamics in the Geoemydidae family. The illustrated phylogeny is consistent with previous cladistics and evolutionary patterns, demonstrating monophyletic grouping of Batagurinae species within the family
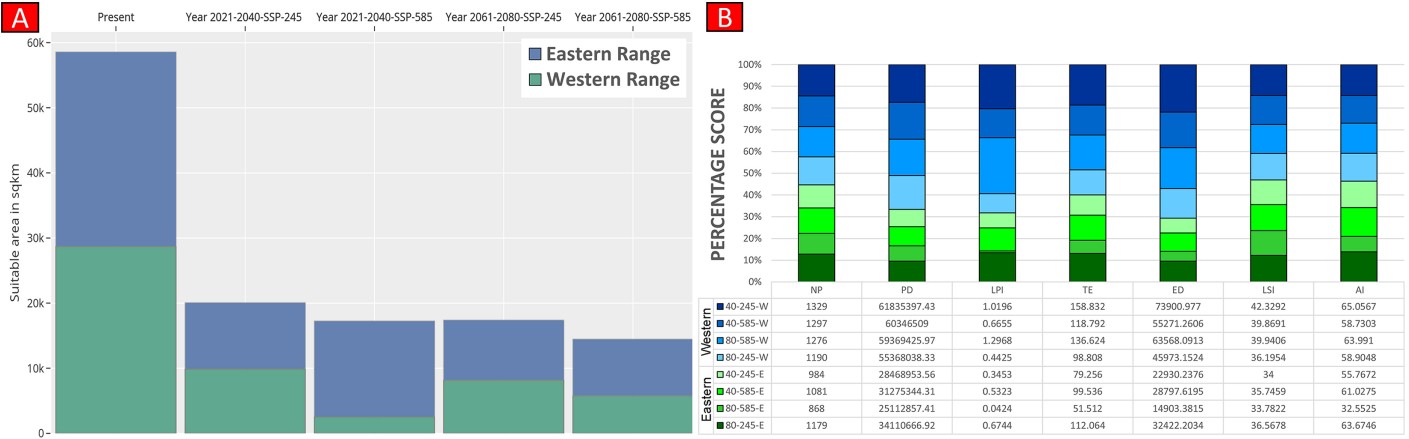

**Figure 8 Habitat quality assessment of *G. hamiltonii* in the western range and eastern range for future climatic scenarios.** Values represent the score of the indices. (A) The suitable habitat for *G. hamiltonii* in present and future climatic scenarios. (B) The percentage stack of class-level matrices applied for habitat quality assessment of *G. hamiltonii* in ssp245 and ssp585 future scenarios for the year between 2021–2040 and 2061–2080 (NP, No. of Patches; PD, Patch Density; LPI, Largest Patch Index; TE, Total Edge; ED, Edge Density; LSI, Landscape Shape Index; AI, Aggregation Index).

Geoemydidae, as well as divergence of *G. hamiltonii* prior to *Paghshura* and *Batagur* species (*Thomson, Spinks & Shaffer, 2021*). However, we believe that further mitogenomic data are needed to determine the real matrilineal connection of this critically endangered turtle group. Further, the structure and variation of the *G. hamiltonii* mitochondrial genome and comparison with other closely related species allowed us to demonstrate their evolutionary relationship. Furthermore, relatively less genetic variation was found between the two mitogenomes of *G. hamiltonii* generated from India (the native range) and China (the non-native range). The similarity of *G. hamiltonii* mitogenomes from two distant localities suggests that illegal trafficking of the species persists, which may have an impact on China's native turtles.

It is evident that the mitogenomic genes (PCGs and CR) have high potential to demonstrate the genetic diversity, potential gene flow, and/or mitochondrial introgression among different Geoemydid populations (*Suzuki & Hikida, 2011*; *Vamberger et al., 2014*; *Ihlow et al., 2016*). Such genetic information is also important for conservation action plans to avoid inbreeding depression, the founder effect, and demographic stochasticity in various reptile species populations (*Harris, Zhang & Nielsen, 2019*; *Kolbe et al., 2012*; *Kundu et al., 2023*). Similarly, the present genetic information will help future population genetics research on *G. hamiltonii* by comparing the nucleotide variations in different mitochondrial genes, particularly PCGs, rRNAs, and CR from different populations. The large-scale population genetic information will further help to better understand and manage the possibly inbred populations of endangered *G. hamiltonii* taxa in India and other countries by accelerating their genetic diversity. This will enable us to make solid inferences on extant species diversity and to deduce recommendations for scientific breeding and reintroduction projects.

The distribution modeling result suggests that the limited species range (7.16%) with fragmented habitat exists in both the eastern and western parts. As most of the habitat

patches within the western range cover the southeastern portion, we suggest prioritizing the identified zone as a conservation priority for *G. hamiltonii* in Pakistan. Further, in the eastern range, habitat patches of *G. hamiltonii* were distributed in the far eastern portion of Assam state, on both edges of the Brahmaputra River, which is heavily influenced by accelerated land-use change (*Pervez & Henebry, 2015*). Thus, we recommend special attention to the biodiversity management authorities in India and Pakistan.

Furthermore, as distance to water bodies and availability were found to positively influence the distribution of *G. hamiltonii*, maintaining the natural environmental flow within the river Brahmaputra should be prioritized to preserve and protect native freshwater biodiversity, including turtles (*Anantha & Bhadbhade, 2018*). Notably, this region also accommodates the highest number of turtle species worldwide (*Buhlmann et al., 2009*). Hence, we suggest that the suitable habitats mapped inside and outside the protected areas may be prioritized to bring them into the protected area network and enhance protection in both ranges through spatial planning for protecting the remaining suitable habitats for this endangered species.

Moreover, the future climate projections help us understand that the massive loss of *G. hamiltonii* habitat (>65%) over the next 50 years is reflected in the impact of climate change on the hydrological regime of the IGB river basins. The IGB river basins are shared by three major river basins, the Indus, the Ganges, and the Brahmaputra, which originate from a large number of glaciated areas in the Himalayan range (*Eriksson et al., 2009*; *Miles et al., 2021*). These Himalayan glaciers are receding faster as a result of rising temperatures induced by greenhouse gas emissions, considerable unpredictability in precipitation trends, and an increase in glacier melt, which will have catastrophic societal and geomorphic impacts on IGB river basins (*Xu et al., 2009*; *Kääb et al., 2012*; *Nepal & Shrestha, 2015*; *Azam et al., 2021*). Such amplified climate changes and spatio-temporal variations have greatly affected the snow cover and surface water areas of the IGB river basins (*Siderius et al., 2013*; *Kiani et al., 2021*; *Mondal et al., 2021*; *Nazeer et al., 2022*; *Uereyen et al., 2022*). In consequence, hydrological extremes, such as floods and droughts, may endanger both human and wildlife habitats in the IGB river basins throughout the 21st century (*Wijngaard et al., 2017*; *Veh, Korup & Walz, 2020*; *Dahri et al., 2021*). Further, river bank erosion and sedimentation, as well as other anthropogenic pressures, operated as significant elements in the changing dynamics of present and future land use and land cover in the IGB river basins (*Collins, Davenport & Stoffel, 2013*; *Caesar et al., 2015*; *Debnath et al., 2022*). It has been demonstrated that in addition to climate change, freshwater megafauna (*e.g.*, the Ganges River dolphin, *Platanista gangetica*, and the Indus River dolphin, *Platanista minor*) are facing increasing pressure from large-scale hydrological changes such as damming and river diversion in the IGB river basins (*Rai et al., 2023*). In this context, ecological pressure should be considered in any hydrological infrastructure development in the IGB river basins, and specific action plans are required to ensure the long-term survival of any vulnerable species.

Due to the advantages of genetic and distributional modeling data, more integrated approaches at the level of different microhabitats levels are needed to design realistic conservation plans for these critically endangered species. Such unified information will

help us identify different populations of target species and facilitate their translocation or reintroduction into preferred wild habitats, reducing interpopulation competition and hybridization as well as conservation risks from the consequences of climate change.

## CONCLUSIONS

Wild populations of spotted pond turtle (*G. hamiltonii*) in South Asian countries are seriously threatened by habitat fragmentation and illegal hunting. Molecular systematics and ecological studies can provide important clues for their proper conservation. Current mitogenomic analyses delineate the evolutionary relationships of *G. hamiltonii* within the family Geoemydidae and recommend the generation of more mitogenomes of Batagurinae representatives to confirm their complete phylogeny. Furthermore, MaxEnt-based species distribution modeling suggests that natural habitat has been greatly affected and reduced by rapidly increasing urbanization. Moreover, the drastic reduction of *G. hamiltonii* habitat over the next 50 years highlights the impact of climate change on the IGB river basins. Therefore, to protect these endangered species in the wild, we highlight the urgent need of proper conservation action plans across their range distribution in South Asian countries.

### Funding

This research was supported by the Basic Science Research Program through the National Research Foundation of Korea (NRF) funded by the Ministry of Education (2021R1A6A1A03039211) and a grant from the Institute of Eminence (IOE), Ref. No./IoE/2021/12/FRP, University of Delhi, India. The Korea Institute of Marine Science and Technology Promotion (KIMST), funded by the Ministry of Oceans and Fisheries (Grant No. 20220214) supported the APC. The funders had no role in study design, data collection and analysis, decision to publish, or preparation of the manuscript.

### Grant Disclosures

The following grant information was disclosed by the authors:
Ministry of Education: 2021R1A6A1A03039211.
Institute of Eminence (IOE): Ref. No./IoE/2021/12/FRP.
Ministry of Oceans and Fisheries: 20220214.

### Competing Interests

The authors declare that they have no competing interests.

### Author Contributions

• Shantanu Kundu conceived and designed the experiments, performed the experiments, analyzed the data, prepared figures and/or tables, authored or reviewed drafts of the article, and approved the final draft.

- Tanoy Mukherjee performed the experiments, analyzed the data, prepared figures and/or tables, authored or reviewed drafts of the article, and approved the final draft.
- Manokaran Kamalakannan performed the experiments, analyzed the data, prepared figures and/or tables, authored or reviewed drafts of the article, and approved the final draft.
- Gaurav Barhadiya performed the experiments, analyzed the data, prepared figures and/or tables, and approved the final draft.
- Chirashree Ghosh performed the experiments, analyzed the data, prepared figures and/or tables, authored or reviewed drafts of the article, project administration and funding acquisition, and approved the final draft.
- Hyun-Woo Kim conceived and designed the experiments, performed the experiments, analyzed the data, prepared figures and/or tables, authored or reviewed drafts of the article, project administration and funding acquisition, and approved the final draft.

### Animal Ethics

The following information was supplied relating to ethical approvals (*i.e.*, approving body and any reference numbers):

No animals were encountered or killed from the wild and did not involve harm to the animal, hence no ethics committee or institutional review board approval is required for the present study. The experimental protocols were approved by the host institutions (Pukyong National University, South Korea; Zoological Survey of India, Indian Statistical Institute, and University of Delhi, India) and all procedures were accomplished in accordance with relevant guidelines and regulations of ARRIVE 2.0. (https://arriveguidelines.org).

### Data Availability

The nucleotide sequence data is available at GenBank: OP344485.

### Supplemental Information

Supplemental information for this article can be found online at http://dx.doi.org/10.7717/peerj.15975#supplemental-information.

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
