# Peer review of "Matrilineal phylogeny and habitat suitability of the endangered spotted pond turtle (Geoclemys hamiltonii; Testudines: Geoemydidae): a two-dimensional approach to forecasting future conservation consequences"

_PeerJ, doi:10.7717/peerj.15975_

## Round 0.1 · original submission · Major Revisions

Dear authors,

After receiving feedback from three reviewers, I believe that the manuscript needs major revisions before it can be accepted. The reviewers have provided comments on certain sections of the methodology as they do not find an integration between the genomic and niche modeling analyses. Additionally, they have noted that some of the conclusions are not supported by the analyses performed.

Sincerely,

Armando Sunny

Reviewer 1 ·

Basic reporting

The manuscript entitled “Matrilineal phylogeny and habitat suitability of the Endangered Spotted Pond Turtle, Geoclemys hamiltonii (Testudines: Geoemydidae): An integrated approach to predicting future conservation impacts by mitogenomics and distribution modelling”, is an important contribution to the evolutionary relationships and current habitat conditions of this endangered species across its native range. In addition, it provides molecular data that can be further used for assessing population genomics patterns.

However, there are critical aspects that should be addressed:

The manuscript structure does not follow the PeerJ standards. Specifically, “Results” and “Discussion” should be considered as separated sections. Moreover, in authors’ “Results and Discussion” section, there is no Discussion in “Model execution and habitat suitability” and “Habitat quality assessment” sub-sections, thus lacking any comparison against previous studies and missing to highlight the valuable findings of this study, which are key components of a research paper.
Regarding the Introduction, there is important information in lines 318-322 that highlight its critical status. I suggest incorporating this information in the Introduction to provide more justification for the conservation status of the species.
Furthermore, a methodological framework that integrates mitogenomics data with species distribution modeling is expected from reading the title. However, there is not such joint analyses and no interpretation from both results (phylogenomic relationships with habitat characteristics).
Therefore, I suggest modifying the title, maybe shortening it just before “An integrated approach…”, and to elaborate a more comprehensive Discussion (as outlined in previous paragraph).

The English language should be improved along the manuscript to avoid misleading understanding. The most relevant case is in line 103, where the usage of “euthanasia” word does not make sense to what is described in the text (i.e. was the animal sacrificed?). Another example is from lines 88-90, it would be easier for the audience to understand if the redaction is changed to something like “Additionally, we conducted Species Distribution Modelling for G. hamiltonii”. More recommendations about alternative words and grammar issues are provided in the attached PDF file (see file attached).

Based on the above, a modification to the title is suggested and considerable improvements to the manuscript are needed.

Experimental design

In line 103: Please correct the word used in “euthanasia”. The correct word should indicate that the organism was sedated, not killed.

In addition, it is not clear how and from where the other G. hamiltonii's mitogenome was generated. Also, two different sequences appear to be used in different analyses: in the phylogenomic reconstruction and in the structural variation. Please, clarify these issues.

Validity of the findings

Discussion of results regarding habitat suitability and quality are missing. Therefore, some conclusions regarding these aspects could have little support. Please, improve this section.

Annotated reviews are not available for download in order to protect the identity of reviewers who chose to remain anonymous.

Reviewer 2 ·

Basic reporting

The manuscript presents sufficiently clear and professional use of the English language. The references of the scientific literature are pertinent for the study, although they could be significantly enriched, especially for the species distribution modeling section, including authors who in recent years have modeled the response of species distribution in future scenarios.

The structure of the article is satisfactory and shows clear figures and appropriate tables. It also provides raw data essential for replication purposes, comprising mitochondrial DNA genomes and geographic occurrences of the studied species. However, the presentation of the results does not exhibit an entirely satisfactory quality despite having interesting and relevant results, particularly due to an insufficient discussion of results, the lack of a precise presentation of hypotheses, and general conclusions that could be significantly enriched with a deeper vision of future impacts by implementing projections of their distribution models.

In addition to this, the manuscript states in its title that the study seeks to be an integrative approach to predict future impacts on the conservation of the endangered species using mitogenomics and distribution modeling, however the phylogenetic relationships of the species with its group is not discussed in more depth, and distribution models are not projected into the future to explore the extent of such impacts. To do so, it would be necessary to carry out such models or reconsider the objectives of this article more cautiously.

If the authors aim to explore future impacts in the habitat suitability of the species I suggest to integrate the following article to the references of an improved version of this manuscript: https://doi.org/10.3390/d11080138 as well as more specialized authors focused on modeling future scenarios for fauna and particularly reptiles, for example:

https://www.ncbi.nlm.nih.gov/pmc/articles/PMC3880584/
https://www.sciencedirect.com/science/article/abs/pii/S0169534711000693
https://onlinelibrary.wiley.com/doi/abs/10.1111/j.1600-0587.2013.00600.x

Otherwise, I recommend not to indicate as an objective 'to predict' future impacts on the distribution patterns of the species, and instead concentrating the study on the findings of its past (phylogeny), and its present (distribution) to glimpse the possible risks in its future.

Experimental design

The contents of the manuscript agree with the aim and focus of the journal, and the research question is well defined and relevant, thus filling an important gap in the information about this species and the taxonomic group to which it belongs. The mitogenomic section of the manuscript presents rigorous and complete research work, however the contents related to species distribution modeling require refinement in the exposition of its parameterization and theoretical foundations, as well as a deeper description of its methods and variables. An example of the latter mentioned is the lack of clarity regarding the variables that were used and under which particular hypotheses.

Validity of the findings

The findings presented in the manuscript are important and I consider that the necessary data for its replicability are available and are clearly exposed throughout the study. However, the conclusions section of the proposal does not satisfactorily expose a solid connection with the results found, and no data or models are presented to support some of the conclusions delivered, particularly in regards to the prediction of future impacts in the conservation of the species.

Reviewer 3 ·

Basic reporting

Throughout the manuscript, the English language should be improved to ensure that an international audience can clearly understand your text. Some examples where the language could be improved are found throughout the revised and attached manuscript. I suggest you have a colleague who is proficient in English and familiar with the subject review your manuscript, or contact a professional editing service.
The article includes sufficient introduction and background to of the topic addressed, and relevant prior literature is appropriately referenced.
The structure of the article it does not conform to an acceptable format of ‘standard sections’ (abstract, introduction, material & methods, results, discussion, conclusions), because it merges the results and the discussion. I believe that it would be more appropriate to separate them, so that the discussion would be much clearer and more understandable. Some examples where the sections (results and discussion) could be improved are found in the revised and attached manuscript. Figures and tables are relevant to the content of the article, of sufficient resolution, and appropriately described and labeled.
The submission is self-contained, represent an appropriate unit of publication, and include all results relevant for the two objectives presented. In this manuscript, there is no hypothesis or research question; i consider that the authors have good information in their introduction, that could raise a hypothesis and/or question.

Experimental design

The manuscript is original primary research and within the Aims and Scope of the journal. The authors do not manage research questions and/or hypotheses in the manuscript, but they do make it clear what their goals are, and identify and contributes the knowledge gap being investigated. I reiterate that the manuscript could be better understand if it had a specific research question or hypothesis.
The research presented is rigorous investigation performed to a high technical standard, but in the ethical standard part I have doubts. The authors mention that the organism used was from captivity and to obtain the sample it was euthanized, but in the manuscript, they never say the reason for this action. There is a concern, they say that it does not require the approval of an ethical committee for not having killed a turtle in the wild, but it is not clear why they killed (euthanasia) an organism in captivity, which is in danger of extinction. I believe that it should be clarified, and consider whether there should be an ethical approval.
The methods described in the manuscript are with sufficient detail & information to replicate by another investigators.

Validity of the findings

The manuscript is not redundant with other works. The data in manuscript on conclusions are robust statistically sound, and limited to supporting results.

Additional comments

In general, the manuscript provides important and relevant information about an endangered turtle. But I do not find the relationship between the data of the metagenomic decoding of Geoclemys hamiltonii and the data of suitability, quality and characterization of the habitat in all its distribution of this species. These two themes in the manuscript seem to be unrelated, I think that in the discussion the authors could have joined these two goals of the work, to make this link clear.
In particular, the paragraphs are extremely long, in the attached revision of the manuscript there are suggestions to avoid such long paragraphs. The figures in the supplementary material that are referenced in the manuscript are not in any order. There are repetitive paragraphs both in the introduction and in the results and discussion section (suggestions are provided in the attached review).

Annotated reviews are not available for download in order to protect the identity of reviewers who chose to remain anonymous.

---

## Round 0.2 · Minor Revisions

Dear authors,

Both reviewers agree that additional minor corrections are still necessary.

I would like to extend my gratitude for your diligent effort in providing the previous observations, and I eagerly await the corrected version.

Best regards,

Armando Sunny

Reviewer 1 ·

Basic reporting

The manuscript now entitled “Matrilineal phylogeny and habitat suitability of the endangered Spotted Pond turtle (Geoclemys hamiltonii; Testudines: Geoemydidae): a two-dimensional approach to forecasting future conservation consequences”, has tremendously improved from its first version. The authors now follow the PeerJ manuscript sections, they incorporate future climatic scenarios modeling and the Discussion of their findings, therefore highlighting not only their results but the main applications for the conservation of this turtle species.

I would like to point out some comments to the main text in a line-number basis:

L 63: Please check the “…” format in “Schedule I”.
L 121: In the first manuscript version, the authors specified the name of the product used for sedation. Please, include it.
L 164: The authors are referring to two web servers. Please, include the links to those servers as it has been done for the previous ones.
L 261-262: There is no mitogenome with a length of 15,505 bp in Table 1 or Table S1. Please, review and correct.
L 297-306: The information contained in the first paragraph and some portion of the second one in the “Major phylogenetic relationship” section, it is not a RESULT. This information should be used in the DISCUSSION section instead, by comparing their findings with previous studies. Please, move and adequate this information properly.
L 310-321: Also, this information should be used in the DISCUSSION section.
L 308: The order of Figure 3 (and Figure 4) at the end of the manuscript, should be checked. This is because in “Figure 3 page”, there is the Figure 4 description and image. Please, review and correct.
L 334: It should be “Figure 5” instead of “Figure 4”. Please, check and correct.
L 336: The same.
L 336: In Figure S3, the description should be corrected as the “T” in “…average sample value. he curves show” statement is missing.
L 350: It is not the “west range”, but the “east range”. Otherwise, it would not be supporting the statement. Please, correct.
L 360: In Figure 8, the color palette and the values arrangement along the bars make difficult the interpretation of this graph. Please, consider a change of colors (maybe a contrasting pattern between West-East ranges or between years).
L 379-380: It is unclear what does “populations of exotic species must pay close attention to genetic diversity screening” mean. Please, review and change redaction.
L 396: Is it really the southernmost portion? Because, “at first sight”, it seems that the southernmost portion of the range is the East, not in the west, also by checking the Latitude.

Experimental design

No comments.

Validity of the findings

No comments.

Reviewer 2 ·

Basic reporting

I have no major comments on the use of language throughout this new version of the manuscript. In the same way, its references have been expanded with pertinent studies that have substantially enriched its methods and discussion. The changes in the structure of the manuscript and the addition of figures 6 and 8 represent a considerable improvement.

Experimental design

The integration of future projections with climate change scenarios based on shared socioeconomic paths promptly addresses the observations raised in the first round of review. In the same way, the methods used are now described in greater detail and clarity, facilitating their replicability.

Validity of the findings

The separation of the results and conclusions contributed to a clearer reading of both sections. Likewise, the latter were notably enriched, allowing a clearer understanding of the research questions with the results obtained and the pertinent scientific literature.

Additional comments

Here are some minor changes and recommendations:

LINE 29. Consider changing 'attempts' for 'aims'.

LINE 30. Consider changing 'recognize' for 'identify' or a synonym.

LINE 46. Consider changing 'approx.' for '≈'

LINE 101. Consider changing 'vital' for 'relevant'

LINES 446-447. The authors point out 'Furthermore, MaxEnt-based species distribution modeling shows that natural
446 habitat has been greatly affected and reduced by rapidly increasing urbanization.', however, the maxent models did not include enough variables or modeling strategies that allow this statement to be made. I recommend changing 'shows' to 'suggests', or rephrasing this particular statement.

LINES 448-449. Consider changing 'we highly demand' for 'we highlight the need' or 'we highlight the urgent need'.

---

## Round 0.3 · accepted · Accept

Dear Authors,

I am delighted to inform you that after careful consideration and multiple rounds of revision, your manuscript has undergone significant improvement, addressing all the revisions requested by the reviewers in a commendable manner. As a result, I am pleased to announce that your work has been accepted for publication.

Best regards,

Armando Sunny.

Reviewer 1 ·

Basic reporting

I have no major comments on the structure, figures, and supplementary material of the manuscript. This version incorporated the suggestions from previous reviews.

Experimental design

No comments.

Validity of the findings

No comments.

Reviewer 2 ·

Basic reporting

No comment.

Experimental design

No comment.

Validity of the findings

No comment.

Additional comments

No comment.